# Understanding Musculoskeletal Loadings among Supermarket Checkout Counter Cashiers: A Biomechanical Analysis

Tânia T. Silva [1], Catarina Sousa [2], Ana Colim [3] and Matilde A. Rodrigues [1,2,3,*]

1   Center for Translational Health and Medical Biotechnology Research, School of Health, Polytechnic Institute of Porto, Rua Dr. António Bernardino de Almeida nº 400, 4200-072 Porto, Portugal; tania4teixeira@hotmail.com
2   Department of Environmental Health, ESS, Polytechnic of Porto, Rua Dr. António Bernardino de Almeida nº 400, 4200-072 Porto, Portugal; catarina.a.m.sousa@hotmail.com
3   Algoritmi Centre, School of Engineering, University of Minho, 4800-058 Guimarães, Portugal; ana.colim@dps.uminho.pt
*   Correspondence: mar@ess.ipp.pt

**Abstract:** Work-related musculoskeletal disorders (WMSD) are highly prevalent among supermarket cashiers. These disorders are frequently related to the adoption of awkward postures and manual materials handling. This study aimed to analyze musculoskeletal loadings in supermarket cashiers, considering the handling of different products and different checkout conditions. To accomplish this, we employed an inertial motion capture system to measure full-body kinematics while simulating 19 cashier tasks. The study included five female cashiers from a supermarket in Northern Portugal, ranging in age from 19 to 61 years old. Using joint angles, material load, and muscle function as input parameters, we conducted the musculoskeletal loadings assessment using the Rapid Upper Limb Assessment (RULA) and Rapid Entire Body Assessment (REBA) methods. Results showed that RULA scores were higher for the microtasks that involved product scanning. Regarding microstasks analyzed by REBA, the replacement of paper rolls for the receipt machine at the checkout counter yields the highest scores. Based on these findings, there is a compelling need to redesign supermarket checkout workstations to alleviate the physical demands placed on cashiers and to ensure organizational sustainability.

**Keywords:** anthropometry database; cashier; ergonomics; food retail; organizational sustainability; work-related musculoskeletal disorders



## 1. Introduction

Work-related musculoskeletal disorders (WMSD) are a high prevalence among supermarket cashiers [1,2]. This is also a public health problem due to the high number of professionals conducting such activities. In fact, in Portugal, in 2021, the food retail sector accounted for 1760 retail food trade units in operation, and employed 84.9 thousand workers [3]. Additionally, there has been a gradual rise in the number of food retail establishments in recent years, with an average annual growth rate of 0.7% between 2018 and 2023 [4].

The cashier is responsible for carrying out a set of functions ranging from reading, weighing, and recording the price of goods, as well as packaging and bagging goods for customer convenience, to receiving and verifying payments [5,6]. Within the range of tasks undertaken by supermarket workers, the responsibilities of cashiers stand out as posing a significant risk of developing WMSD. According to Lehman et al. [7], cashiers are among the top 10 occupations with a heightened susceptibility to these disorders.

Body regions most susceptible to injury include the neck, upper limbs (shoulders, elbows, hands/wrists), and lower back [6,8–10]. A study developed in Portugal revealed prevalent musculoskeletal complaints among hypermarket cashiers, including non-specific

pain, cervical, and lumbar pain, primarily affecting the shoulder, cervical spine, and lumbar spine [11]. Additionally, findings showed that neck and back pain significantly hindered cashiers from performing their regular tasks, prompting them to seek medical attention from nurses, doctors, or physiotherapists. Some even had to modify their job responsibilities and reduce their activities at home due to the discomfort caused by severe pain [6].

The work of a cashier is typically performed at a checkout counter. In this workstation, tasks are characterized by repetitive and monotonous movements of the upper limbs, as well as manual handling of heavy and/or bulky loads [12]. This, related to the poor checkout design, contributes to musculoskeletal disorders in the neck, shoulders, and wrists [13]. To address the physical strain caused by sitting and standing, the workstation is designed for the cashier to primarily work in a standing position, with periodic breaks to sit [13–15]. This helps to prevent muscle fatigue in the lower limbs during the shift and allows the cashier to alternate positions and rest different muscle groups [15,16]. However, to fulfill their job responsibilities, cashiers frequently engage in forward flexion, leaning sideways, and trunk rotation, placing them at risk for developing back pain, as noted by Maciukiewicz et al. [10] and Rodacki et al. [12]. Consequently, checkout cashiers commonly experience postural issues due to the inherent strain in their work, which is further exacerbated by improper biomechanics in their workstations [12].

Despite some checkout counters being replaced by smart solutions, like automated and intelligent systems, such as self-checkout counters, that are reducing their presence in some supermarkets, the work of cashiers is still considered relevant, and their role will not disappear. They serve as the company's frontline representatives, often being the initial and final point of contact for many customers. Due to their substantial responsibilities and integral role in the organization, they are highly regarded as essential and trusted workers [17,18]. Given their pivotal role, it is imperative to enhance their working conditions to ensure the continuity of their job duties and safeguard their health and safety. Therefore, it is critical to better understand the musculoskeletal loadings that cashiers are exposed. Postural analysis holds significant potential as an effective technique for assessing work activities. The assessment of the loads on the musculoskeletal system of supermarket cashiers, considering posture, muscle function and the forces they exert, can contribute to better characterizing the biomechanical component of the risk related to the development of WMSD and can considerably contribute to the implementation of necessary changes. Therefore, having access to observational methods, such as Rapid Entire Body Assessment (REBA) [19] and Rapid Upper Limb Assessment (RULA) [20] is advantageous for ergonomists according to Hignett and McAtamney [19].

This study aimed to understand how checkout design can influence musculoskeletal loadings, characterizing its implications among cashiers with different anthropometric characteristics and dominant hands, as well as among checkouts with different designs and product features.

## 2. Materials and Methods

### 2.1. Subjects

The study was conducted in one retail company located in Northern Portugal. To this end, a supermarket with almost 400 workers was selected. The existence of two different checkouts and their size was one of the criteria for this choice. The two existing checkouts were the most representative ones in the company's supermarkets across the country. The supermarket under study was the one with the highest number of checkout operators, totaling 102 cashiers.

This study included 5 cashiers, i.e., those who voluntarily agreed to participate in this study and who met the inclusion criteria: Portuguese nationality and female gender. To ensure compatibility with the motion capture system, participants were excluded if their BMI was higher than or equal to $30 \text{ kg/m}^2$, since the system's larger shirt sizes were not suitable for these subjects.

Through a brief interview, relevant information was gathered from each participant. Subjects exhibited variations in anthropometric measures, dominant hand, and work experience (Table 1): three participants were older ($\bar{x}$ = 56 years old; sd = 6.2) while the other two were younger ($\bar{x}$ = 21 years old; sd = 2.82), two had shorter stature $\bar{x}$ = 155 cm; sd = 2.82), and three were taller ($\bar{x}$ = 166.3 cm; sd = 5.1). Additionally, one participant was ambidextrous, another left-handed, and the remaining three were right-handed. The three older participants showed signs of musculoskeletal pain, but we chose to retain them and not consider musculoskeletal symptoms as an exclusion factor.

**Table 1.** Study participants characteristics.

| Participant | Height (cm) | Age (Years) | Work (Years) | Work Weekly (Hours) | Dominant Hand |
|---|---|---|---|---|---|
| Participant 1 | 172 | 23 | 4 | 40 | Right |
| Participant 2 | 153 | 49 | 29 | 30 | Right |
| Participant 3 | 165 | 58 | 34 | 40 | Both |
| Participant 4 | 162 | 61 | 31 | 30 | Left |
| Participant 5 | 157 | 19 | 0.33 | 16 | Right |

The study was approved by the Ethics Committee of the School of Health of the Polytechnic Institute of Porto (CE0061D).

*2.2. Motion and Postural Analysis*

2.2.1. Workstation and Tasks

To conduct ergonomic analyses, each participant was evaluated at two checkout workstations, where cashiers could adopt a standing or sitting posture. Dimensions of the checkout counter and the corresponding reaches of the participants are summarized in Table 2.

**Table 2.** Checkout counter dimensions.

| Area | Height (cm) | Width (cm) | Depth (cm) | Reach (of the Worker) (cm) |
|---|---|---|---|---|
| Counter | 86 | - | 50 | - |
| Touch-screen monitor | 31 | 29 | - | 36 |
| Scanner | 20 | - | - | 48 |
| Money drawer | 10 | 46 | 15.5 | - |
| Thrash can | 40 | - | 52 | - |
| Shelf 1 | 52.2 | 50 | 44.4 | - |
| Shelf 2 | 27 | 50 | 44.4 | - |
| Shelf 3 | 3 | 50 | 44.4 | - |
| Shelf 4 | 49 | 30.5 | 24.5 | - |
| Shelf 5 | 3 | 44.5 | 24.5 | - |
| Scale | 4.5 | - | - | - |
| Scale counter | 83.5 | - | - | - |
| Support counter | 83 | - | - | - |
| Receipt printer | 6.3 | - | - | - |
| Telephone | 6.3 | - | - | - |
| Magazine Display | - | 50 | - | 87.3 |

The participants simulated 19 microtasks frequently performed in this working section (Table 3). The microtasks were discerned through careful observation of the workstation and interviews with the personnel. The checkout counters were equipped with registers arranged from left to right. The primary distinction between the two evaluated checkout stations is the presence of an additional conveyor belt. In one of the analyzed counters, this conveyor belt is sliding, facilitating the delivery of items to the customer, while the other counter, does not possess this sliding feature. Figure 1 portrays (a) the checkout counter both without and (b) with the conveyor belt.

**Table 3.** Processes, macrotasks, and microtasks performed at the checkout counter.

| Process | Macrotask | Microtask |
|---|---|---|
| Storage | Treatment of values | Carrying the cash drawer to the checkout counter |
| Storage | Exhibition and replacement | Carrying paper rolls for the receipt machine to the checkout counter |
| Storage | Exhibition and replacement | Replacing paper rolls for the receipt machine at the checkout counter |
| Storage | Customer services | Collecting customer baskets and transport them to the basket storage area |
| Storage | Treatment of values | Carrying the cash drawer and other materials to the vault room |
| Sales | Customer services | Opening the checkout counter opening |
| Sales | Customer services | Closing the checkout counter opening |
| Sales | Customer services | Picking up and delivering the client's card and delivering the receipt |
| Sales | Customer services | Scanning bulky/heavy items while standing (with a conveyor belt) |
| Sales | Customer services | Scanning bulky/heavy items while standing (without a conveyor belt) |
| Sales | Customer services | Scanning light products while standing (with a conveyor belt) |
| Sales | Customer services | Scanning light products while standing (without a conveyor belt) |
| Sales | Customer services | Scanning light products while sitting (with a conveyor belt) |
| Sales | Customer services | Scanning light products while sitting (without a conveyor belt) |
| Sales | Customer services | Weighing fruits and vegetables |
| Cleaning | Cleaning of the section, equipment, and utensils | Cleaning the checkout counter (with a conveyor belt) |
| Cleaning | Cleaning of the section, equipment, and utensils | Cleaning the checkout counter (without a conveyor belt) |
| Support | Treatment of values | Placing money in safety bags and sealing them |
| Support | Treatment of values | Placing the money safety bags in the safe |

### 2.2.2. Instruments and Measurement Protocol

Kinematic analysis was conducted through the XSens MVN whole-body motion capture system (XSens technologies, Enschede, The Netherlands) which comprises 17 inertial measurement units (IMU) sensors that tracked the motion data during the predefined tasks. This system provides data about 3D joint angles, the center of body mass, as well as temporal parameters, such as segment position, which facilitates gait analysis.

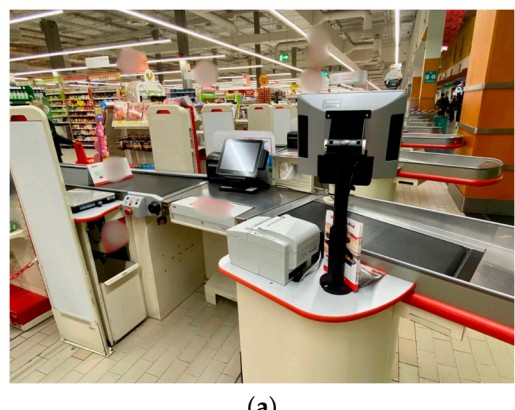 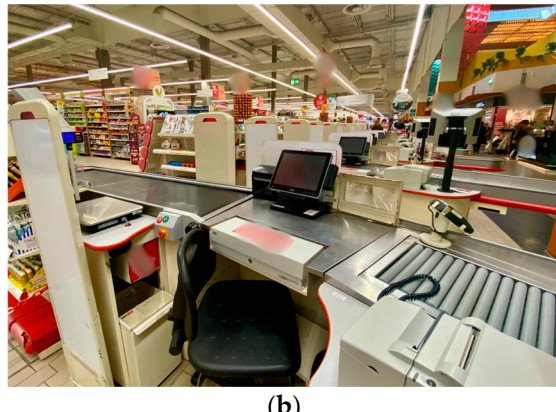

(**a**)                                                                (**b**)

**Figure 1.** (**a**) Checkout counter without conveyor belt; (**b**) checkout counter with conveyor belt.

For each participant, anthropometric data were collected, encompassing measurements such as stature and shoe length. These collected data points were subsequently employed to construct the MVN human model utilized using XSens.

Following the placement of IMU on the subjects' body landmarks, calibration procedures were meticulously executed, strictly adhering to the previously established protocol, which was developed with consideration of previous studies, e.g., [21,22], and the guidelines outlined in the XSens Manual [23]. These procedures involved having the participant assume the N-pose and subsequently perform a walking trial.

Subsequently, each participant completed the 19 simulated microtasks, reflective of the tasks in this section. Each assigned microtask was systematically undertaken three times per participant to ensure data consistency and reliability. Microtask assessments were initially conducted at one of the checkout workstations, without a conveyor belt. Subsequently, for microtasks involving "scanning products" and "cleaning the checkout counter" which necessitated evaluation at a separate checkout station, participants were relocated to an alternative workstation equipped with a conveyor belt. "Scanning products" was conducted using an integrated scanner. The registration of products followed a left-to-right sequence where the cashiers used, normally, one hand to grab the lightweight products and both hands to grab the heavier/bulkier ones. Both hands were employed to pass the product through the scanner, and, if needed, they collaboratively moved and rotated the item to locate the barcode.

In Figure 2, two study participants and their avatars can be observed engaged in distinct tasks. On the left, there is scanning of bulky/heavy products while standing in a checkout counter with a conveyor belt, while on the right, scanning lightweight products is depicted while sitting in a checkout counter without a conveyor belt.

Regarding the microtask that involved scanning products, common purchase items were chosen. We included both lightweight and heavier or bulkier products. The lightweight items (<5 kg) chosen included: 1 bag of rice, 1 can of tuna, 1 pack of Portuguese bread (5 units), 1 bottle of shampoo, 1 pack of pasta, 1 cheese ball, 1 can of sausages, 1 pack of cherry tomatoes, 1 razor blade box (with an alarm), 1 pack of flour, and 1 pack of 4 liquid yogurts. The chosen heavier or bulkier products (≥5 kg) included: 1 pack of 6 water bottles (1.5 L each), 1 codfish, 1 dustbin, 1 pack of potatoes (5 kg), 1 pack of diapers (144 units), 1 baby car seat (with an alarm), 1 pack of beer bottles (24 units), 1 clothes drying rack, 1 liquid laundry detergent, 1 bottle of water (6 L), 1 pack of 6 milk cartons (1 L each), 1 pack of wood pellets (15 kg), 1 pack of wood (10 kg), and 1 pack of dry dog food (20 kg). Concerning the microtask of "weighing fruits and vegetables", the selected items were 1 courgette, 4 oranges, and 1 papaya.

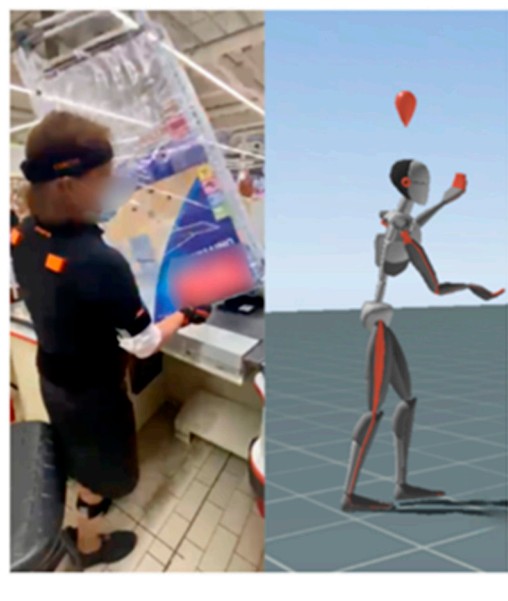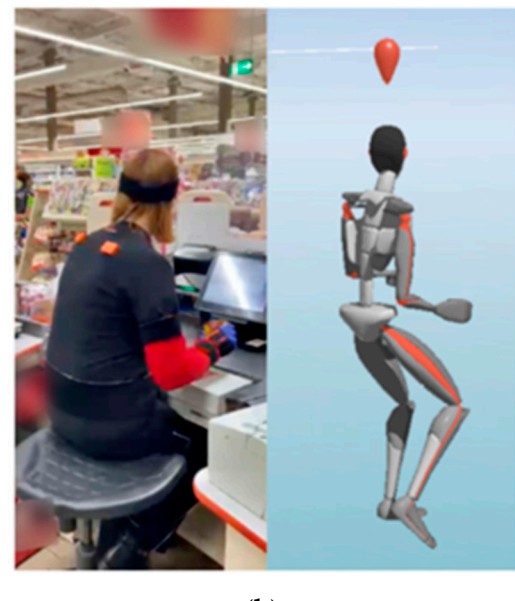

(**a**)　　　　　　　　　　　　　　　　　　　　　　　(**b**)

**Figure 2.** Study participants and their corresponding XSens outputs: (**a**) scanning heavy items while standing at a checkout counter with a conveyor belt; (**b**) scanning light products while sitting at a checkout counter without a conveyor belt.

The motion capture system, XSens MVN, was employed to track various aspects such as orientation, position, movement, and center of mass across different body parts. Subsequently, the collected data were sent wirelessly to a computer equipped with software capable of observing, recording, and analyzing the movements.

The raw data were initially collected and processed using XSens MVN software version 2021 (XSens technologies, Enschede, The Netherlands). This analysis entailed the use of graphical representations depicting joint angles, as well as the assessment of movement speed and duration. Subsequently, REBA and RULA values were computed. This application provided a semi-automatic ergonomic assessment, integrating the motion data collected using the IMU and manually entering "muscle" and "force" values for RULA, and "load/force", "coupling", and "activity" values for REBA. These values were determined considering the observations completed in loco, as well as the load of the product. The final REBA and RULA scores were subsequently presented in tabular form. To analyze these final scores, Tables 4 and 5 present the REBA and RULA action levels, respectively. Tasks involving the entire body were assessed using the REBA method, while those exclusively engaging the upper body were assessed using RULA.

**Table 4.** REBA action levels (Adapted from Hignett & McAtamney [19]).

| Action Level | REBA Score | Risk Level | Action (Including Further Assessment) |
| :---: | :---: | :---: | :---: |
| 0 | 1 | Negligible | None necessary |
| 1 | 2–3 | Low | May be necessary |
| 2 | 4–7 | Medium | Necessary |
| 3 | 8–10 | High | Necessary soon |
| 4 | 11–15 | Very high | Necessary now |

**Table 5.** RULA action levels (Adapted from McAtamney & Corlett [20]).

| Action Level | RULA Score | Risk Level |
|:---:|:---:|:---:|
| 1 | 1–2 | Acceptable |
| 2 | 3–4 | Further investigation is required and changes may be required |
| 3 | 5–6 | Investigation and changes are required soon |
| 4 | 7 | Investigation and changes are required immediately |

## 3. Results

Table 6 presents the RULA and REBA scores derived from the analysis of microtasks conducted at the cashier station. Results denoted the high loads on the musculoskeletal system of cashiers. However, RULA and REBA scores differ according to the microtask under analysis. Tasks requiring the engagement of the entire body (evaluated using REBA) were found to be less detrimental to the assessed cashiers than tasks exclusively involving the upper body (assessed using RULA). However, some of the tasks that required all body movements were classified with high-loading levels. The microtask "Replacing paper rolls in the receipt machine at the checkout counter" demonstrated a propensity for causing physical strain on cashiers, with REBA scores ranging from 7 to 11 (scores varying based on the participant under analysis). This microtask often necessitates cashiers to assume precarious positions, occasionally requiring them to kneel on the floor or adopt a squatting posture. Notably, the participant who exhibited the highest REBA score for this microtask was participant 1, who was taller and younger and only had a few years of work experience.

Table 6 highlights the microtasks that pose the greatest loadings to cashiers, notably "Scanning products", which received a RULA score of 7, signifying the maximum scoring level for all the assessed participants. This microtask is characterized by continuous wrist rotation, elevated shoulders, and repetitive movements of the upper limbs. When comparing the two checkout counters, no noticeable differences were detected regarding the presence of the conveyor belt. Regarding the selected products, despite conducting a distinct evaluation between lighter and heavier/bulkier items, the resulting values remained equally high.

The evaluation of "Scanning bulky/heavy items, while sitting" was not conducted given the fact that cashiers typically perform this microtask while standing, as it is more convenient and less physically demanding to handle and maneuver products in that posture.

Another critical microtask is "Weighing Fruits and Vegetables", with RULA scores of 5, 6, and 7 (scores varying based on the participant under analysis). In this activity, cashiers must rotate their trunks to the left to access the scale, further contributing to the ergonomic challenges faced by the cashiers.

Microtasks that appeared to pose lower loadings included "Carrying the cash drawer to the checkout counter", "Carrying the cash drawer and other materials to the vault room", and "Cleaning the checkout counter", as indicated by REBA scores of 4, 5, and 6.

**Table 6.** Postural analysis, through REBA and RULA in the checkout station.

| Process | Macrotask | Microtask | Method | Participant | | | | | | | | | |
|---|---|---|---|---|---|---|---|---|---|---|---|---|---|
| | | | | P1 | | P2 | | P3 | | P4 | | P5 | |
| Storage | Treatment of values | Carrying the cash drawer to the checkout counter | REBA | 6 | | 4 | | 4 | | 4 | | 5 | |
| Storage | Exhibition and replacement | Carrying paper rolls for the receipt machine to the checkout counter | REBA | 5 | | 5 | | 4 | | 7 | | 4 | |
| Storage | Exhibition and replacement | Replacing paper rolls for the receipt machine at the checkout counter | REBA | 11 | | 7 | | 8 | | 9 | | 9 | |
| Storage | Customer services | Collecting customer baskets and transport them to the basket storage area | REBA | 6 | | 6 | | 6 | | 6 | | 5 | |
| Storage | Treatment of values | Carrying the cash drawer and other materials to the vault room | REBA | 6 | | 5 | | 5 | | 4 | | 4 | |
| Sales | Customer services | Opening the checkout counter opening | REBA | 5 | | 4 | | 4 | | 4 | | 4 | |
| Sales | Customer services | Closing the checkout counter opening | REBA | 7 | | 5 | | 4 | | 5 | | 4 | |
| Sales | Customer services | Picking up and delivering the client's card and delivering the receipt | RULA (L/R) | 5 | 5 | 4 | 4 | 4 | 4 | 6 | 6 | 4 | 5 |
| Sales | Customer services | Scanning bulky/heavy items while standing (with a conveyor belt) | RULA (L/R) | 7 | 7 | 7 | 7 | 7 | 7 | 7 | 7 | 7 | 7 |
| Sales | Customer services | Scanning bulky/heavy items while standing (without a conveyor belt) | RULA (L/R) | 7 | 7 | 7 | 7 | 7 | 7 | 7 | 7 | 7 | 7 |
| Sales | Customer services | Scanning light products while standing (with a conveyor belt) | RULA (L/R) | 7 | 7 | 7 | 7 | 7 | 7 | 7 | 7 | 7 | 7 |
| Sales | Customer services | Scanning light products while standing (without a conveyor belt) | RULA (L/R) | 7 | 7 | 7 | 7 | 7 | 7 | 7 | 7 | 7 | 7 |
| Sales | Customer services | Scanning light products while sitting (with a conveyor belt) | RULA (L/R) | 7 | 7 | 7 | 7 | 7 | 7 | 7 | 7 | 7 | 7 |
| Sales | Customer services | Scanning light products while sitting (without a conveyor belt) | RULA (L/R) | 7 | 7 | 7 | 7 | 7 | 7 | 7 | 7 | 7 | 7 |
| Sales | Customer services | Weighing fruits and vegetables | RULA (L/R) | 7 | 7 | 5 | 5 | 6 | 5 | 7 | 7 | 6 | 7 |
| Cleaning | Cleaning of the section, equipment, and utensils | Cleaning the checkout counter (with a conveyor belt) | REBA | 6 | | 5 | | 4 | | 5 | | 4 | |
| Cleaning | Cleaning of the section, equipment, and utensils | Cleaning the checkout counter (without a conveyor belt) | REBA | 5 | | 5 | | 5 | | 5 | | 4 | |
| Support | Treatment of values | Placing money in safety bags and sealing them | RULA (L/R) | 6 | 6 | 5 | 7 | 3 | 5 | 6 | 6 | 4 | 5 |
| Support | Treatment of values | Placing the money safety bags in the safe | RULA (L/R) | 4 | 4 | 7 | 7 | 6 | 7 | 7 | 6 | 4 | 4 |

Note: RULA (L/R) = RULA (Left/Right); P1 = Participant 1; P2 = Participant 2; P3 = Participant 3; P4 = Participant 4; P5 = Participant 5.

## 4. Discussion

This study confirms substantial musculoskeletal strain in cashiers, notably in tasks like scanning and weighing, aligning with the existing literature that associates such strain with the development of musculoskeletal disorders [12,19,20]. These findings were already expected, since numerous studies have consistently reported a high prevalence of musculoskeletal symptoms among grocery cashiers [8,10].

Higher musculoskeletal loading levels were found for microtasks related to customer service, in particular, the ones of scanning and weighing. These tasks are related to repetitive movements, manual material handling, rotation, and lateral bending of the trunk, as well as difficult reaches [12]. This, along with insufficient rest and long journeys, is related to musculoskeletal discomfort in the shoulders, neck, and lower back [10].

In terms of differences among cashiers, a comprehensive analysis unveiled that Participant 1, who was the tallest, youngest, and right-handed, had the highest REBA and RULA values along the different microtasks under analysis. Closely behind was Participant 4, who was left-handed, on the taller side, and older. Concerning hand dominance, Participant 3 was ambidextrous, Participant 4 was left-handed, and the other 3 participants were right-handed. However, this variable demonstrated no influence on the analyzed microtasks, according to the reported values of REBA and RULA.

In this domain, the interaction with the cashiers revealed that older cashiers with longer work experience commonly reported more musculoskeletal complaints, consistent with the existing literature suggesting that these workers are susceptible to musculoskeletal pain and disorders [6,24].

It was observed that working in a standing and in a sitting posture conceded consistent RULA assessments for the microtasks involving product scanning. As previously mentioned, these assessments consistently reached the maximum RULA score of 7, indicating that changes in the work environment are required immediately. Relatively to both postures (standing and sitting) in the scanning and weighing microtasks, the results obtained were not expected, since some findings indicate that a standing position offers biomechanical advantages for the upper limbs and trunk [7,16]. Nevertheless, the RULA score obtained for these microtasks was consistently high, making it challenging to differentiate between the two postures, both of which were deemed as posing a risk. However, it is important to be aware that is advisable to incorporate a combination of both standing and sitting for optimal working conditions [7,16]. This approach mitigates the risk of lower limb muscle fatigue throughout the work shift, underscoring the importance of having chairs available at each checkout station [7,16]. In addition, exploring other alternatives such as introducing new chair designs that allow for an intermediate posture between sitting and standing could provide significant benefits, particularly for individuals experiencing discomfort during prolonged standing. Research in this area is warranted. For instance, a recent study by Noguchi et al. [25] introduced a new chair design, and participants reported a notable reduction in lower leg and lower back discomfort compared to traditional standing positions. Furthermore, the implementation of scheduled rest breaks is essential for cashiers, even given the demanding nature of their work, to safeguard their well-being [16].

Our findings also demonstrate that taller cashiers presented higher musculoskeletal loadings. This emphasizes the importance of customizing workstations to accommodate the individual anthropometric characteristics of cashiers. Considering this, is imperative that the checkout counter is height-adjustable to guarantee ergonomic and comfortable working postures, as emphasized in the study by Lang et al. [26]. It is also recommended to investigate the effects of rotating cashiers between mirrored configurations to assess whether this strategy has the potential to mitigate the overloading of one arm [26].

In the present study, no differences in scores between different checkout conditions were observed. This was because the maximum scoring level was achieved in both, not allowing to distinguish between both designs. This suggests that new design solutions are needed for checkouts.

One contemporary challenge arises from the increasing prevalence of self-checkout stations in supermarkets. Nonetheless, it is crucial to underscore that the significance of cashiers' roles cannot be overstated. However, the frequent errors and the expectation for customers to assume a more active role at self-checkout counters have resulted in many customers favoring the traditional checkout counter with a cashier, where the checkout process is typically smoother and more reliable [27]. Additionally, cashiers function as the primary face of the company, interacting with a multitude of customers daily. The enduring importance of this job role highlights the necessity of continuously enhance working conditions, ensuring the seamless execution of their duties while prioritizing their health and safety [27].

Applying ergonomic principles to the design processes, workplace, and organizational structure serves not only as a response to legal requirements, but also as a strategic alignment with companies' objectives. Hence, checkout counters should be redesigned with meticulous attention to biomechanical and anthropometric principles [28].

The results underscore the crucial role of ergonomic checkout counter design, emphasizing the need to tailor it to individual characteristics and advocate for a balanced combination of standing and sitting (as suggested by Lehman et al. [7], Draicchio et al. [16], and Cudlip et al. [29]).

Despite the relevance of this study's results, it is important to realize that only five cashiers were included in the assessments. These were the workers who voluntarily accepted to be a part of the study and who met the inclusion criteria. To mitigate this effect, a significant number of microtasks were assessed. However, in the future, it will be relevant to better understand the influence of certain variables, including larger comparative samples, such as dominant arm and gender. Additionally, the way the product is held, whether with one or two hands, may influence the results.

The study was limited to the analysis of the biomechanical component of the risk of WMSD in cashiers. Future studies should address other variables, such as psychosocial and organizational risk factors, that also play a relevant role in the risk of WMSD in checkout operators [9]. Continuous efforts to enhance working conditions, even in the era of self-checkout, are crucial for the health and safety of cashiers.

Additionally, despite the fact that results were limited to the design of the checkouts under analysis, we consider the analyzed checkouts representative of the retail reality in the country. Different results could be obtained in other checkout conditions and with a larger sample.

## 5. Conclusions

The study emphasizes the significant and negative impact of musculoskeletal loads on cashiers, significantly contributing to the risk of WMSD development. Additionally, it underscores the importance of designing checkout counters respecting ergonomic requirements, considering individual characteristics, and promoting a balanced combination of standing and sitting postures. To significantly reduce the risk of developing these disorders, the checkout counter should offer adjustability in all directions to accommodate 95% of the cashier population. In cases where this is not feasible, the height and reach dimensions should be determined based on the tallest individuals followed by the provision of equipment to adjust the height to accommodate smaller individuals. The workstation should enable cashiers to adopt various safe working postures during task performance while keeping their joints in neutral positions. Task rotation and appropriate resting times are also relevant to prevent WMSD among these professionals.

Experienced and older cashiers frequently reported musculoskeletal complaints, consistent with the literature indicating that cashier workers are prone to musculoskeletal pain and injuries.

The importance of future studies lies in evaluating the compatibility between the dimensions of supermarket checkout counters and the anthropometric measurements of workers. Redesigning supermarket checkouts with meticulous consideration of workers'

anthropometric dimensions is vital to alleviate the physical demands of the job and ensure organizational sustainability. This is crucial to enable older individuals in the retail sector to continue working in favorable conditions and reduce their susceptibility to musculoskeletal injuries. Additionally, it is important to ensure optimal working conditions for newcomers in the retail sector as cashiers, preventing the occurrence of musculoskeletal injuries.

Despite the emergence of self-checkout counters, traditional checkout stations will persist, making it essential to prioritize the safety and health of these operators.

**Author Contributions:** T.T.S.: conceptualization, methodology, investigation, statistical analysis and writing; M.A.R.: supervision, conceptualization, writing—review and validation; A.C.: supervision, conceptualization, writing—review and validation; C.S.: investigation, statistical analysis and writing. All authors have read and agreed to the published version of the manuscript.

**Funding:** This research received external funding under the project MSD@RETAIL and supported by FCT—Fundação para a Ciência e Tecnologia within the R&D Units Project Scope: UIDB/00319/2020.

**Institutional Review Board Statement:** The study was conducted according to the guidelines of the Declaration of Helsinki, and the study was approved by the Committee of Ethics of the School of Health of the Polytechnic Institute of Porto (approval number CE0061D).

**Informed Consent Statement:** Written informed consent was obtained from all subjects involved in the study.

**Data Availability Statement:** Data are contained within the article.

**Conflicts of Interest:** The authors declare no conflicts of interest.

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
