# Peer review of "Understanding Musculoskeletal Loadings among Supermarket Checkout Counter Cashiers: A Biomechanical Analysis"

_safety, 2023_

Round 1

Reviewer 1 Report

Comments and Suggestions for Authors

Dear authors,

The topic addressed is of interest but, in my opinion, there are some critical aspects that should be reviewed in order to consider the work for publication.

General observations:

-          I suggest you reconsider the use of the terms 'risk' and 'risk assessment' in the text. Technically, the risk of WMSD is the probability of developing a disorder in a group of workers exposed to specific risk factors and should not be confused with a measure of exposure to a specific biomechanical risk factor. WMSDs have a multifactorial etiopathological mechanism in which the role of psychosocial and other non-biomechanical risk factors may be important. Knowledge about the relationship between the level of biomechanical exposure, including cumulative exposure, and the likelihood of onset of the disorders is limited.

-          In general, reading your paper, it is not clear to the reader whether or not the workstation layout and the tasks analysed are representative of the work patterns for this category of workers in your country and compared to what is described in the literature. This aspect is important in order to understand whether and to what extent your results are generalisable.  I suggest you improve this aspect: if the situation described is generalisable, it could enhance your results.

Punctual observations:

1.       (line 66) what do you mean by "intelligent solution"? It is not clear, explain it.

2.       (lines 87-94)  The criteria for including participants are unclear. 5 cashiers out of 400? Why? How many cashiers could be included?  Age group, presence/absence of illnesses?

3.       (lines 95-99) The table of anthropometric measurements should be clearer.

4.       You should add how you collected personal data from participants (e.g. questionnaire/interview...) and what kind of data you collected (age, data, illness...).

5.       Figure 1: not very clear, could you improve the quality? (perhaps a more detailed single figure would be better)

6.       Figure 3 and Figure 4 can be briefly described in the text and possibly included as supplementary material

7.       (line 241) check reference 16 (this is not an epidemiological study on the prevalence of MSDs)

8.       (line 239) I suggest being more cautious: you do not have data on the prevalence of MSDs in the selected population (e.g., data from occupational health surveillance), so you could only claim that your results support the existence of a high level of postural risk factors for MSDs.

(lines 284-287) I suggest reviewing the critical analysis of the limitations in more detail, highlighting how the limitations have been handled and/or how they could be overcome.

Author Response

Dear Reviewer,

Thank you for dedicating time to review our manuscript and for providing insightful comments that have significantly contributed to its improvement.

Please find our reply in the enclosed document.

Best regards,

Matilde Rodrigues

Reviewer 2 Report

Comments and Suggestions for Authors

Thank you for allowing me to be the first to read the article titled "Understanding the risk of work-related musculoskeletal disorders among checkout counter cashiers: A biomechanical analysis." It is clear that there is a high prevalence of musculoskeletal disorders in the retail sector, and XSense and its associated MVN software are undoubtedly suitable for assessing these risks.

However, I find the current research methodology presentation not up to the scientific standard; instead, it seems more like an introductory step to a regular development program. The article overlooks that a new cash register design has emerged and spread in the meantime, based on a thorough ergonomic analysis. Therefore, the referenced literature should be approached with caution.

We must learn about individual workstations' characteristics, dimensions, distances, and layout in the methodology description. We have yet to learn the details of the only five participants whose knowledge becomes significant later. 

It needs to be clear how the activity analysis presented in Table 2 was validated. 

Based on the description, each individual performed the tasks only once, so the measured times' validity is unreliable. The differences in the subjects' work methods must be captured and interpreted. How the work is done needs to be revealed, for example, whether product barcode scanning is done with a built-in or manual scanner. Methods to determine force and grip values in rows 167-177 must be clarified. It is unclear why the analysis was not conducted using REBA and RULA methods for all the tasks, as comparing REBA and RULA scores is unfounded. 

Discussion-worthy text appears several times among the results, especially in rows 225-237. 

The recommendation in rows 260-263 regarding a standing workstation seems awkward, as manual material handling restrictions based on a calculated lifting mass limitation can eliminate the movement of heavy goods, making the cashier's workstation convertible to a sitting position. Adjustable seat height and a footrest may be sufficient for this.

While interpreting the results, it is necessary to elaborate on the other activities in the job and their effects. 

The figures could be more precise to interpret.

Again,  the article contains important information, but due to the mentioned shortcomings, it requires further refinement and accuracy. I hope my comments assist in the revision and improvement of the manuscript.

Author Response

(The authors gave the same response as above.)

Reviewer 3 Report

Comments and Suggestions for Authors

The authors conducted an interesting study, the aim of which was to understand how the design of supermarket checkouts may shape the risk of WMSD and to determine the extent to which this risk varies among employees with different anthropometric characteristics and dominant hand, as well as the importance of different cash resources. record product designs and features.

A well-planned study was conducted during which body posture was objectively assessed while performing 19 activities typical of cashier work at two cash registers (with and without a conveyor belt). Body posture was determined using the XSens MVN whole-body motion capture system (XSens Technologies, Enschede, The Netherlands), which consists of 17 inertial measurement unit (IMU) sensors that tracked movement data during predefined tasks. The study was conducted among 5 women with different anthropometric characteristics. Ergonomic assessment of body posture while performing selected activities was carried out using two methods: Rapid Entire Body Assessment (REBA) and Rapid Upper Limb Assessment (RULA), which allow predicting the risk of WMSD. Using this procedure, it was possible to identify the most burdensome activities and indicate the introduction of necessary changes.

However, I would suggest emphasizing that determining the amount of load while performing individual activities is not sufficient to predict the risk of pathology in a specific employee, and especially in the entire professional group. It is necessary to take into account how often individual activities are performed during work, the pace of work, the working time, and the number and distribution of breaks during work. However, the presented procedure is very suitable for identifying the most strenuous activities and those elements of the workplace that require ergonomic intervention.

Author Response

Dear Reviewer,

Thank you for dedicating time to review our manuscript and for providing insightful comments that have significantly contributed to its improvement.

Please find our reply bellow.

Best regards,

Matilde Rodrigues

The authors conducted an interesting study, the aim of which was to understand how the design of supermarket checkouts may shape the risk of WMSD and to determine the extent to which this risk varies among employees with different anthropometric characteristics and dominant hand, as well as the importance of different cash resources. record product designs and features.

A well-planned study was conducted during which body posture was objectively assessed while performing 19 activities typical of cashier work at two cash registers (with and without a conveyor belt). Body posture was determined using the XSens MVN whole-body motion capture system (XSens Technologies, Enschede, The Netherlands), which consists of 17 inertial measurement unit (IMU) sensors that tracked movement data during predefined tasks. The study was conducted among 5 women with different anthropometric characteristics. Ergonomic assessment of body posture while performing selected activities was carried out using two methods: Rapid Entire Body Assessment (REBA) and Rapid Upper Limb Assessment (RULA), which allow predicting the risk of WMSD. Using this procedure, it was possible to identify the most burdensome activities and indicate the introduction of necessary changes.

Thank you for the appreciation of our work. We are pleased that you recognize the importance of the study and the obtained results.

I would suggest emphasizing that determining the amount of load while performing individual activities is not sufficient to predict the risk of pathology in a specific employee, and especially in the entire professional group. It is necessary to take into account how often individual activities are performed during work, the pace of work, the working time, and the number and distribution of breaks during work. However, the presented procedure is very suitable for identifying the most strenuous activities and those elements of the workplace that require ergonomic intervention.

We agree and additional explanations about the role of other risk factors were added.

Round 2

Reviewer 1 Report

Comments and Suggestions for Authors

Dear Authors,

I greatly appreciate the revision work done which has significantly improved the manuscript. Having no further comments, I recommend that the editor accept it in its current form for publication.

Author Response

We are grateful for your positive feedback on the manuscript revisions. Your recommendation for acceptance is appreciated and your invaluable input has been crucial.

Reviewer 2 Report

Comments and Suggestions for Authors

Thank you for the revised version of your paper, which has significantly improved the clarity of your research presentation. The modifications made in response to our previous review are appreciated. However, a few concerns still exist that need attention. 

While your paper effectively presents a thorough workplace assessment, the discussion section must be stronger. For example, in lines 107-108, you mention existing back pains and differences between workers with or without back pains. It is crucial to delve deeper into these aspects, providing a more comprehensive analysis and contextualizing your findings within the existing literature. Extend the discussion beyond the workplace assessment, incorporating new findings, scientific results, and practical applications. These changes will add depth and relevance to your research.

The conclusion needs enrichment with messages of scientific interest. Summarize the key findings and their implications for the field. Discuss how your research contributes to the existing body of knowledge and suggest potential avenues for future research. By doing so, you will enhance the overall impact of your study.

Mind, you have two pages with number 8 on them. 

I appreciate your efforts and look forward to the enhanced version of your valuable contribution.

Comments on the Quality of English Language

Small grammatical enhancements will contribute to the overall professionalism of the paper.

Author Response

Revisor 2

Thank you for the revised version of your paper, which has significantly improved the clarity of your research presentation. The modifications made in response to our previous review are appreciated. However, a few concerns still exist that need attention. 

Thank you for your feedback. We are glad to hear that the revisions have improved the clarity. We appreciate your acknowledgement of the modifications.

While your paper effectively presents a thorough workplace assessment, the discussion section must be stronger. For example, in lines 107-108, you mention existing back pains and differences between workers with or without back pains. It is crucial to delve deeper into these aspects, providing a more comprehensive analysis and contextualizing your findings within the existing literature. Extend the discussion beyond the workplace assessment, incorporating new findings, scientific results, and practical applications. These changes will add depth and relevance to your research.

Thank you for your guidance. We enhanced the discussion section (lines 268-271; 275-278; 284-290; 318-321;). Your feedback was valuable, and we appreciate the opportunity to improve the paper.

The conclusion needs enrichment with messages of scientific interest. Summarize the key findings and their implications for the field. Discuss how your research contributes to the existing body of knowledge and suggest potential avenues for future research. By doing so, you will enhance the overall impact of your study.

Thank you for your constructive feedback. We worked on enriching the conclusion by summarizing key findings, highlighting their implications, and emphasizing the contribution to the existing body of knowledge (lines 338-342; 350-352). We also included suggestions for potential avenues of future research to enhance the overall impact of the study (lines 353-363). Your guidance is appreciated.

Mind, you have two pages with number 8 on them

Thank you for noticing that. We rectify the duplicate page numbers promptly.

Reviewer 3 Report

Comments and Suggestions for Authors

The authors corrected and supplemented the text in accordance with the reviewers' suggestions. I have no further comments and suggest that the article be published without further changes.

Author Response

Thank you for your encouraging words. We are glad the revisions met your expectations. Your support is greatly appreciated throughout this process.

Round 3

Reviewer 2 Report

Comments and Suggestions for Authors

Dear Authors,

Thank you for considering my suggestions. In my standard, you made it.

Bests